# The Use of Electronic Medical Records-Based Big-Data Informatics to Describe ALT Elevations Higher than 1000 IU/L in Patients with or without Hepatitis B Virus Infection

**DOI:** 10.3390/v13112216

**Published:** 2021-11-04

**Authors:** Hiroyuki Amano, Tatsuo Kanda, Hitoshi Mochizuki, Yuichiro Kojima, Yoji Suzuki, Kenji Hosoda, Hiroshi Ashizawa, Yuko Miura, Shotaro Tsunoda, Yosuke Hirotsu, Hiroshi Ohyama, Naoya Kato, Mitsuhiko Moriyama, Shuntaro Obi, Masao Omata

**Affiliations:** 1Department of Gastroenterology, Yamanashi Central Hospital, 1-1-1 Fujimi, Kofu, Yamanashi 400-8506, Japan; hiroyuki_amano613@yahoo.co.jp (H.A.); jynnhitoshi@gmail.com (H.M.); y-kojima@ych.pref.yamanashi.jp (Y.K.); y-suzuki2a@ych.pref.yamanashi.jp (Y.S.); hosoda-anba@ych.pref.yamanashi.jp (K.H.); zero.in.on.1231@gmail.com (H.A.); miuray@yamanashi.ac.jp (Y.M.); tsunoshou.3.6.11@gmail.com (S.T.); ohyama.hiroshi@chiba-u.jp (H.O.); momata-tky@umin.ac.jp (M.O.); 2Division of Gastroenterology and Hepatology, Department of Medicine, Nihon University School of Medicine, 30-1 Oyaguchi-kamicho, Itabashi-ku, Tokyo 173-8610, Japan; Moriyama.mitsuhiko@nihon-u.ac.jp; 3Genome Analysis Center, Yamanashi Central Hospital, 1-1-1 Fujimi, Kofu, Yamanashi 400-8506, Japan; hirotsu-bdyu@ych.pref.yamanashi.jp; 4Department of Gastroenterology, Chiba University, Graduate School of Medicine, 1-8-1 Inohana, Chuo-ku, Chiba 260-8670, Japan; kato.naoya@chiba-u.jp; 5Department of Internal Medicine, Teikyo University Chiba Medical Center, 3426-3 Anesaki, Ichihara 299-0111, Chiba, Japan; obishun@gmail.com; 6The University of Tokyo, 7-3-1 Hongo, Bunkyo-ku, Tokyo 113-0033, Japan

**Keywords:** ACLF, acute liver failure, ALT elevation, HBV reactivation, HBV surface antigen

## Abstract

Hepatitis B virus (HBV) infection is one of the serious health problems in the world as HBV causes severe liver diseases. Moreover, HBV reactivation has occasionally been observed in patients with resolved HBV infection and patients using immunosuppression and anticancer drugs. Large-scale hospital data focused on HBV infection and severe liver function were analyzed at our hospital, located in an urban area adjacent to Tokyo, the capital city of Japan. A total of 99,932 individuals whose blood samples were taken at 7,170,240 opportunities were analyzed. The HBV surface antigen (HBsAg)-positive group had a more frequent prevalence of patients with higher transaminase elevations than the HBsAg-negative group. However, among the HBsAg-negative group, patients who were positive for anti-HBV surface antibody and/or anti-HBV core antibody, had more severe liver conditions and fatal outcomes. More careful attention should be paid to alanine transaminase (ALT) elevations higher than 1000 IU/L in patients who had current and previous HBV infection.

## 1. Introduction

Hepatitis B virus (HBV) infection causes acute hepatitis, acute liver failure, chronic hepatitis, cirrhosis and hepatocellular carcinoma (HCC) [1]. There is a long history of studies about occult HBV infection in patients without HBV surface antigen (HBsAg), but with anti-HBV surface antibody (anti-HBs) and/or anti-HBV core antibody (anti-HBc) [2]. In general, occult HBV infection indicates previous HBV infection. Among patients with resolved HBV infection, attention should also be paid to HBV reactivation, as there are a lot of studies about HBV reactivation related to severe liver diseases including life-threatening hepatitis [3,4]. However, there are few studies, based on large-scale hospital data, about HBV infection and severe liver diseases.

It is important to discover the incidence and risk factors for HBV reactivation [5,6,7]. Few studies have focused on the analysis of large-scale hospital data, especially those focusing on HBV infection and severe liver dysfunction. Various tools have recently been reported to analyze laboratory data by data mining [8,9,10]. Artificial intelligence in this area may also make us better at both performing data analysis and in clinical practice.

After 20 years of HBV vaccination, 85–90% of those vaccinated have protective anti-HBs levels [11]. In Japan, although anti-HBV vaccination has been recommended for health care workers, family members who live with HBV-infected individual and newborns from HBV-infected mother, universal vaccination programs for children against HBV infection just started in October 2016 [12]. These facts mean that, among almost all Japanese patients, anti-HBs positivity indicates a previous HBV infection.

In the present study, we analyzed large-scale hospital data and focused on HBV infection and severe liver function at our hospital. A total of 99,932 individuals who had blood samples taken at 7,170,240 opportunities were analyzed. We revealed the frequency of occurrence and severity of HBV infection in patients with higher transaminase elevations. The HBV surface antigen (HBsAg)-positive group showed a more frequent prevalence of patients with transaminase elevations higher than the HBsAg-negative group. Among the HBsAg-negative group, patients who were anti-HBsAg antibody (anti-HBs)-positive had severe liver conditions and fatal outcomes, although the incidence of severe conditions was low (~1%). Our results may support the previous research reporting that HBV reactivation could lead to severe conditions even among patients with resolved HBV infection.

## 2. Patients and Methods

### 2.1. Patients and Data Collection

Big-data of our all-patients’ cohort seen during a ~10-year period (1 January 2008–28 February 2018) were obtained from the electronic medical records of our hospital, which is a 650-bed medical center at Kofu, Yamanashi, Japan. These big data (totaling 1.2 TB) were deposited into data warehouse (DWH) and analyzed using in house Unix Shell & R (version 3.6.3, Vienna, Austria) script [13,14,15].

During those 10 years, blood samples were taken from 99,932 individuals (Figure 1). Fourteen parameters described below were measured at a total of 7,170,240 instances.

The seven biochemical parameters included: (1. aspartate transaminase (AST), 2. alanine transaminase (ALT), 3. γ-glutamyl transferase (γ-GTP), 4. albumin, 5. total bilirubin, 6. hemoglobin, 7. platelet counts). Six HBV markers were also recorded: ((1) HBsAg, (2) anti-HBs, (3) anti-HBV core antibody (anti-HBc), (4) HBe antigen (HBeAg), (5) anti-HBeAg antibody (anti-HBe) and (6) HBV DNA) and 1 hepatitis C virus (HCV) marker (anti-HCV antibody (anti-HCV). Baseline data were defined at the nearest point to the date which viral markers were obtained.

Of them, 64,992 patients were confirmed for HBsAg status. Further, we divided the patients into three groups as follows: 1. HBsAg-positive; 2. HBsAg-negative but anti-HBs and/or anti-HBc-positive; and 3. triple-negative (negative for HBsAg, anti-HBs, or anti-HBc) groups (Figure 1). A total of 8387 individuals were analyzed in the present study (Figure 1).

### 2.2. Clinical and Laboratory Assessments

Laboratory data and clinical pictures of patients were obtained from the data warehouse. Hematological and biochemical tests including HBV markers were measured by standard laboratory techniques at Yamanashi Central Hospital central laboratories. In the present study, we sampled and analyzed laboratory data at the nearest date confirmed for viral markers, as a baseline.

### 2.3. Measurement of HBV DNA Levels

The HBV DNA level was measured by a commercially available real-time PCR-based assay: transcription-mediated amplification (TMA) assay, COBAS Amplicor HBV Monitor assay or COBAS TaqMan (Roche Diagnostics, Branchburg, NJ, USA) [16]. We converted the HBV DNA levels using the following formula: (log IU/mL) = (log copies/mL) − 0.76 (http://www.jsh.or.jp/member/archives/21, accessed on 1 November 2016). HBV DNA equal to or more than 2.1 log IU/mL was defined as HBV DNA detectable, and HBV DNA less than 2.1 log IU/mL was defined as HBV DNA undetectable in the present study.

### 2.4. Definition of Highly Suspected HBV Individuals

The criteria of liver injury due to HBV was defined as follows; 1. HBV DNA levels increased, or 2. other etiologies, such as drug-induced and ischemic liver injuries, and autoimmune liver diseases, could be excluded.

### 2.5. Ethics

This study protocol was approved by the institutional review board at Yamanashi Central Hospital (No. 29-19), in accordance with the 1964 Helsinki declaration and its later amendments or comparable ethical standards. Details of participation in the study were posted at Yamanashi Central Hospital.

### 2.6. Statistical Analysis

Data are expressed as the means ± standard deviations (SDs). Statistical analyses were performed by univariate analysis with Student’s t-test or chi-squared test; *p* < 0.05 was considered statistically significant. Statistical analysis was performed with DA Stats software version PAF01644 (NIFTY Corp., Tokyo, Japan).

## 3. Results

### 3.1. Patient Characteristics of Three Groups (HBsAg-Positive, HBsAg-Negative but Positive for Anti-HBs/Anti-HBc, and Triple-Negative Groups) Classified by Three Principal HBV Markers

First, we examined how often the HBV and HCV viral markers were measured in 73,634 and 72,700 individuals, respectively. In total, 811 (1.1%) of 73,379; 2369 (25.7%) of 9214; 2128 (28.6%) of 7428; 78 (10.0%) of 781; 463 (60.0%) of 772 and 2959 (4.1%) of 72,700 patients were positive for HBsAg, anti-HBs, anti-HBc, HBeAg, anti-HBe and anti-HCV, respectively.

We categorized patients whose serum was tested for three HBV markers (HBsAg, HBsAb and HBcAb) into three groups, namely (1) HBsAg-positive (*n* = 811), (2) HBsAg-negative and anti-HBs-positive and/or anti-HBc-positive (*n* = 2959) and (3) triple-negative group (*n* = 4617) (Figure 1).

The characteristics of the three groups are shown in Table 1. The HBsAg-positive group is male-dominant, compared with the HBsAg-negative and anti-HBs-positive and/or anti-HBc-positive group (*p* = 0.00519) or triple-negative group (*p* = 0.00396). The individuals in the HBsAg-negative and anti-HBs-positive and/or anti-HBc-positive groups are younger than those in the HBsAg-positive group (*p* < 0.001) or triple-negative group (*p* < 0.001). The individuals in the HBsAg-positive group are older than those in the triple-negative group (*p* = 0.0201).

The AST and ALT levels of the HBsAg-positive group are higher than those of the HBsAg-negative and anti-HBs-positive and/or anti-HBc-positive groups (*p* < 0.001 and *p* < 0.001, respectively). The AST and ALT levels of the triple-negative group are higher than those of the HBsAg-negative and anti-HBs-positive and/or anti-HBc-positive groups (*p* < 0.001 and *p* < 0.001, respectively). The γ-GTP, albumin, platelet counts and hemoglobin levels are significantly different among each group (Table 1).

### 3.2. Highly Suspected HBV-Causing ALT Elevations Higher than 1000 IU/L among HBsAg-Positive Patients

A total of 24 (3.0%) of 811 patients in the HBsAg-positive group had higher ALT elevations. The cause of the higher ALT elevations was HBV, which was highly suspected in 22 (91.7%) of the 24 patients (Table 2). Among highly suspected HBV individuals with higher ALT elevations, liver failure and fatal outcomes were observed in 2 (9.1%) and 2 (9.1%) of 22 patients in this group, respectively. That is, 20 (90.9%) of 22 patients were fully recovered (Table 2).

The cause of higher ALT elevations was suspected as non-HBV-related in 2 (8.3%) of 24 HBsAg-positive patients with higher ALT elevations. Cardiac arrest and heart failure were causes of higher ALT elevations in two non-HBV suspected patients in the HBsAg-positive group (a 64-year male and 79-year male) (Table 3).

Together, 4 (18.2%) of 22 patients who were highly suspected of having HBV-causing ALT elevations higher than 1000 IU/L, had severe hepatitis associated with HBV infection in this group. Again, only 2 (9.1%) of these 22 patients died due to HBV-related hepatic events (Table 2).

### 3.3. Highly Suspected HBV-Causing ALT Elevations Higher than 1000 IU/L among HBsAg-Negative, but Positive for Anti-HBs/Anti-HBc Patients

A total of 30 (1.0%) of 2959 patients in the anti-HBs-positive group had higher ALT elevations. The cause of the higher ALT elevations was HBV, which was highly suspected HBV in 7 (23.3%) of the 30 patients (Table 4). In all seven cases, anti-HBc was positive. Among highly suspected HBV individuals with higher ALT elevations, liver failure and fatal outcomes were observed in two (28.6%) and three (42.9%) of seven patients in this group, respectively. Finally, three patients (42.9%) died (Table 4).

The cause of higher ALT elevations was suspected to be non-HBV related in 23 (76.7%) of 30 anti-HBs-positive patients with higher ALT elevations. Concerning the non-HBV causes of higher ALT elevations among the 23 anti-HBs-positive patients, ischemic liver injury, drug-induced liver injury, obstructive jaundice, acute hepatitis caused by non-HBV, acute hepatitis type C and unknown etiology were the causes in 13 (56.5%), 3 (13.0%), 2 (8.7%), 2 (8.7%), 1 (4.3%) and 2 patients (8.7%), respectively (Table 5).

Of note, five (71.4%) of seven patients who were highly suspected of having HBV-causing ALT elevations higher than 1000 IU/L, had severe hepatitis associated with HBV infection in this group. Again, 3 (42.9%) of these 7 patients died due to HBV-related hepatic events (Table 4).

### 3.4. Highly Suspected Non-HBV-Causing ALT Elevations Higher than 1000 IU/L among Triple-Negative Patients

A total of 75 (1.6%) of 4, 617 triple negative individuals presented higher ALT elevations. Concerning the non-HBV causes of higher ALT elevations among the 75 triple negative patients, ischemic liver injury, drug-induced liver injury, autoimmune hepatitis, obstructive jaundice, post hepatectomy, alcoholic cirrhosis, Epstein–Barr virus (EBV) hepatitis, cytomegalovirus (CMV) hepatitis, idiopathic thrombocytopenic purpura, hepatitis C, hepatitis A, traumatic liver injury and hemophagocytic syndrome were the causes in 24 (32.0%), 22 (29.3%), 6 (8.0%), 4 (5.3%), 4 (5.3%), 5 (6.7%), 3 (4.0%), 2 (2.7%), 1 (1.3%), 1 (1.3%), 1 (1.3%), 1 (1.3%) and 1 patient (1.3%), respectively (Table 5). There are no cases of highly suspected HBV-causing ALT elevations higher than 1000 IU/L among this group.

## 4. Discussion

The present study established that chronic HBV infection or resolved HBV infection is a risk factor for a higher ALT elevation (more than 1000 IU/L) or death associated with higher ALT elevations in association with highly suspected HBV infection. Of interest, although the prevalence of higher ALT elevation was 24/811 (29.6%) or 30/2959 (1.0%), the prevalence of death associated with highly suspected HBV infection was 2/22 (9.1%) or 3/7 (42.9%) in the HBsAg-positive and anti-HBs-positive individuals, respectively. In triple-negative groups, the prevalence of higher ALT elevations “with association with suspected non-HBV infection” was 75/4617 (1.6%). These results strongly suggest that attention should be paid to patients with resolved HBV infection as well as current HBV infection in daily clinical practice. Yet, even if HBsAg was positive, ~70% of those would not have higher ALT elevation.

It has been reported that there is a correlation between the titer of anti-HBc and the presence of viral antigens in the liver [17]. Omata et al. reported that HBcAg was observed in the liver of anti-HBs/anti-HBc positive patients [4]. There has been an ongoing debate about occult HBV infection for a long time [18,19]. In the present study, electronic medical records-based big-data informatics confirmed the significant importance of occult HBV infection in the association with severe life-threatening liver disease in anti-HBs positive individuals, despite its prevalence being relatively low.

In the present study, we also observed HBsAg-negative patients with acute hepatitis and fatal outcomes (Table 4). Although HBsAg-negative individuals positive for anti-HBc and/or anti-HBs are usually defined as patients with resolved HBV infection, HBV DNA reappeared in 15–33% of these patients [20]. HBV reactivation in patients with resolved HBV infection also seemed to be involved in patients with liver failure and death among the anti-HBs-positive individuals (Table 4). It is useful for these patients to be treated with nucleo(s)tide analogs, as these drugs may improve their conditions in patients of this group.

In Japan, nucleo(s)tide analogs have been used for the treatment of hepatitis B infection. Therefore, it is possible that some HBsAg positive patients might receive treatment with nucleo(s)tide analogs. When Fujita et al. studied HBV reactivation in 76,641 patients receiving treatment for rheumatoid arthritis from the Japanese National Database of Health Insurance Claims and Specific Health Checkups data between 2013 and 2016, HBsAg, anti-HBs and anti-HBc were examined in 21,633 (28.2%), 9593 (12.5%) and 11,213 (14.6%) patients, respectively, and nucleo(s)tide analogs were given in 59/76,641 (0.076%) subjects at baseline [6]. In the present study, none had taken nucleos(t)ide analogues at higher ALT elevations than 1000 IU/L among HBsAg-positive group (Table 2) or HBsAg-negative and anti-HBs and/or anti-HBc-positive group (Table 4).

Technologies have progressed to allow the analysis of HBV infection using big data. Lee et al. analyzed a nationwide long-term cohort study using Taiwan’s National Health Insurance Research Database to obtain data on 185,843 patients chronically infected with HBV and demonstrated that the use of nucleos(t)ide analogues for HBV infection was significantly associated with a reduction in intrahepatic cholangiocarcinoma risk [21].

Several studies demonstrated that daily aspirin treatment may be associated with a reduced risk of HBV-related HCC occurrence and recurrence, and liver-related mortality [22,23]. Thus, it is useful to use big data in the analysis HBV infection to establish new findings regarding the association between HBV infection and other diseases [24,25].

There have been few studies, based on large-scale hospital data about HBV infection and severe liver diseases. We also applied electronic medical records-based big-data informatics to characterize higher ALT elevations in 99,932 individuals with or without hepatitis B virus infection. As the present study is a retrospective analysis at a single institute, further prospective studies with larger cohorts will be needed.

HBV reactivation associated with the use of immunosuppressive treatment remains one of the major causes of liver related deaths in the Asia-Pacific region, where HBV infection is endemic [26]. In the present study, we observed HBV reactivation associated with the use of immunosuppressive treatment in two patients (Table 2 and Table 4). Attention should be paid to patients with resolved HBV infection during the immunosuppressive treatment (Table 4). It seems important to screen all patients for HBV infection before the beginning of immunosuppressive treatment and administer pre-emptive nucleos(t)ide analogues to some patients with a risk of HBV reactivation [26,27,28].

There are several limitations in this study. In the present study, we observed ALT serious elevations among HBsAg-positive patients more often than those among the other groups. As it has been difficult to continuously follow-up each patient for a long period of time in only one hospital, we could not calculate annual incident rates of HBV-causing ALT elevations among each group. The sensitivity of the measurement of HBV markers, including HBsAg and HBV DNA, has been changed [29]. Still, our study demonstrated real-world data in our hospital [27,30]. Further study may be needed.

## 5. Conclusions

The present study discloses the frequency and severity of HBV infection in patients with higher ALT elevations. In the anti-HBs-positive group, although the incidence of severe cases was low, some cases with severe liver dysfunction became fatal. We may evaluate higher ALT elevations in anti-HBs-positive patients more carefully.

## Figures and Tables

**Figure 1 viruses-13-02216-f001:**
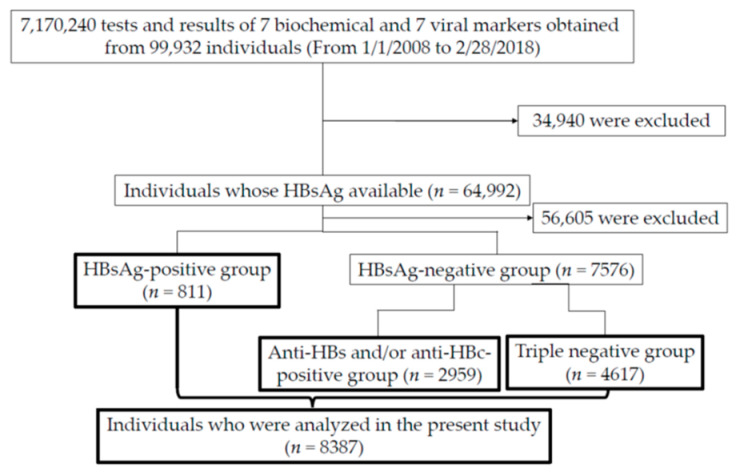
Flowchart demonstrating the inclusion individuals and 3 groups (HBsAg-positive, negative but positive for anti-HBs/anti-HBc and triple-negative groups) in the present study. In the HBsAg-positive, negative but positive for anti-HBs/anti-HBc and triple-negative groups, blood sampling were performed at 20.3 times/person (*n* = 16,489), 22.4 times/person (*n* = 66,388) and 30.2 times/person (*n* = 139,302), respectively. HBsAg, HBV surface antigen; anti-HBs, anti-HBs antibody; anti-HBc, anti-HBV core antibody; triple-negative group, group including individuals negative for HBsAg, anti-HBs or anti-HBc.

**Table 1 viruses-13-02216-t001:** Patient characteristics of 3 groups (HBsAg-positive, HBsAg-negative but positive for anti-HBs/anti-HBc and triple-negative groups).

Patient Groups	HBsAg-Positive	HBsAg-Negative but Anti-HBs and/or Anti-HBc-Positive	Triple-Negative	*p*-Values
Number	811	2959	4617	
Sex (male/female)	458/353	1505/1454	2351/2266	NS
Age (years)	59.6 ± 16.1	55.6 ± 21.6	58.1 ± 17.1	0.0201
AST (IU/L)	75.6 ± 255.3	41.0 ± 80.4	58.9 ± 264.7	NS
ALT (IU/L)	74.4 ± 342.4	38.8 ± 93.4	57.5 ± 220.8	NS
γ-GTP (IU/L)	47.7 ± 76.1	73.0 ± 189.5	86.7 ± 180.7	0.00159
Albumin (g/dL)	4.1 ± 0.6	3.9 ± 0.7	4.0 ± 0.7	<0.001
Total bilirubin (mg/dL)	1.1 ± 1.7	1.1 ± 1.8	1.1 ± 1.7	NS
Platelets (×104/μL)	19.9 ± 9.0	21.5 ± 9.9	23.0 ± 11.5	<0.001
Hemoglobin (g/dL)	13.3 ± 2.0	13.0 ± 2.1	12.8 ± 2.4	<0.001

HBsAg, hepatitis B virus (HBV) surface antigen; anti-HBs, anti-HBV surface antibody; anti-HBc, anti-HBV core antibody; NS, not significant difference.

**Table 2 viruses-13-02216-t002:** Highly suspected hepatitis B virus (HBV) patients with HBV surface antigen (HBsAg)-positivity and alanine transaminase (ALT) elevations higher than 1000 IU/L.

Case	Age (Years)/Sex	Type of Diseases	Outcomes/NUC	HBV DNA (LIU/mL)	Anti-HBs/ Anti-HBc	HBeAg/ Anti-HBe
1	21/male	Acute hepatitis	Recovery/ETV (1.0)	5.5	−/+	+/−
2	23/female	Acute hepatitis	Recovery/None	5.9	−/+	+/−
3	25/male	Acute hepatitis	Recovery/ETV (1.0)	7.5	−/+	+/+
4	28/male	Acute hepatitis	Recovery/ETV (0.5)	8.2	−/+	+/−
5	28/female	Acute hepatitis	Recovery/ETV (0.5)	8.2	−/+	−/−
6	28/male	Acute hepatitis	Recovery/None	6.0	−/+	+/−
7	31/male	Acute hepatitis	Recovery/None	7	−/+	+/+
8	33/male	Acute hepatitis	Recovery/None	6.2	−/+	−/+
9	35/male	Acute hepatitis	Recovery/ETV (0.5)	5.4	−/+	+/−
10	36/male	Acute hepatitis	Recovery/ETV (0.5)	9.1	−/+	+/−
11	45/male	Liver failure	Deceased/ETV (1.0)	7.5	−/+	−/+
12	49/male	Acute hepatitis	Recovery/ETV (0.5)	6	−/+	+/−
13	50/male	Acute hepatitis	Recovery/ETV (1.0)	2.5	−/+	−/+
14	51/male	Acute hepatitis	Recovery/None	6.5	+/+	+/−
15	51/male	Acute hepatitis	Recovery/ETV (0.5)	4.4	−/+	−/+
16	52/male	Acute hepatitis	Recovery/Unknown	2.5	−/+	+/−
17	54/male	Acute hepatitis	Recovery/None	5.6	−/+	−/+
18	54/male	Acute hepatitis	Recovery/None	6.0	−/+	+/−
19	55/male	Liver failure	Deceased/ETV (0.5)	6.7	−/+	−/+
20	56/male	Acute hepatitis	Recovery/ETV (0.5)	4.2	−/+	+/−
21	64/male	Acute hepatitis	Recovery/ETV (0.5)	5.4	−/+	NA/NA
22	65/male	Acute hepatitis	Recovery/Unknown	2.1	−/+	−/+

Only one patient (case 5) had taken steroid treatment for 7 years and took 15 mg daily prednisolone when ALT elevation was observed. Others did not receive any immunosuppressive treatment. No patients had any nucleos(t)ide analogues (NUCs) at ALT elevations. But only one patient (case 11) had NUCs before ALT elevation, and we cannot rule out this as a cause of his ALT elevation. NUC, treatment of NUC after ALT elevations; ETV (0.5), 0.5 mg daily entecavir; ETV (1.0), 1.0 mg daily entecavir; anti-HBs, anti-HBV surface antibody; anti-HBc, anti-HBV core antibody; HBeAg, HBe antigen; anti-HBe, anti-HBe antigen antibody; −, negative; +, positive; NA, not available.

**Table 3 viruses-13-02216-t003:** Non-hepatitis B virus (HBV) suspected patients who had alanine transaminase (ALT) elevations higher than 1000 IU/L among the HBV surface antigen (HBsAg)-positive group.

Case	AGE (Years)/Sex	Type of Diseases	Outcomes/ NUC	HBsAg/HBV DNA (LIU/mL)	Anti-HBs/ Anti-HBc
23	64/male	Ischemic liver injury	Recovery/None	+/NA	NA/NA
24	79/male	Ischemic liver injury	Recovery/None	+/NA	NA/−

None had received immunosuppressive treatment or had taken nucleos(t)ide analogues (NUCs) at ALT elevations. NUC, treatment of NUC after ALT elevations; anti-HBs, anti-HBV surface antibody; anti-HBc, anti-HBV core antibody; −, negative; +, positive; NA, not available.

**Table 4 viruses-13-02216-t004:** Highly suspected hepatitis B virus (HBV) patients with HBV surface antigen (HBsAg)-negativity but anti-HBV surface antibody (anti-HBs)/anti-HBV core antibody (anti-HBc)-positivity and alanine transaminase (ALT) elevations higher than 1000 IU/L.

Case	Age (Years)/Sex	Type of Diseases	Outcomes/NUC	Anti-HBs/ Anti-HBc	HBeAg/ Anti-HBe
25	52/male	Acute hepatitis	Recovery/None	+/+	NA/NA
26	55/male	Liver failure	Deceased/None	+/+	NA/NA
27	65/male	Acute hepatitis	Recovery/None	NA/+	NA
28	70/male	Acute hepatitis	Deceased/ETV (0.5)	+/+	NA
29	76/male	Acute hepatitis	Unknown/None	+/+	+/+
30	81/female	Acute hepatitis	Recovery/None	NA/+	NA/NA
31	82/male	Liver failure	Deceased/None	−/+	NA/NA

Only one patient (Case 28) had received rituximab- cyclophosphamide/hydroxydaunorubicin/oncovin/prednisolone (R-CHOP) for 0.5 years. Others did not receive any immunosuppressive treatment when ALT elevation was observed. No patients had any nucleos(t)ide analogues (NUCs) at ALT elevations. NUC, treatment of NUC after ALT elevations; ETV (0.5), 0.5 mg daily entecavir; HBeAg, HBe antigen; anti-HBe, anti-HBe antigen antibody; −, negative; +, positive; NA, not available.

**Table 5 viruses-13-02216-t005:** Various types of liver diseases causing non-hepatitis B virus (HBV) related higher alanine transaminase (ALT) elevations than 1000 IU/L among the HBV surface antigen (HBsAg)-negative but anti-HBV surface antibody (anti-HBs)/anti-HBV core antibody (anti-HBc)-positive group and triple negative group.

	Number of Patients among the HBsAg-Negative but Anti-HBs/Anti-HBc-Positive Group (*n* = 23)	Number of Patients among Triple Negative Group (*n* = 75)
Ischemic liver injury	13	24
Drug-induced liver injury	3	22
Autoimmune hepatitis	0	6
Obstructive jaundice	2	4
Post hepatectomy	0	4
Alcoholic cirrhosis	0	5
Acute viral hepatitis caused by non-HBV	3	7
Idiopathic thrombocytopenic purpura	0	1
Traumatic liver injury	0	1
Hemophagocytic syndrome	0	1
Unknown etiology	2	0

NA, not available.

## Data Availability

All data underlying this article are available in this article.

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
