# Peer review of "The Use of Electronic Medical Records-Based Big-Data Informatics to Describe ALT Elevations Higher than 1000 IU/L in Patients with or without Hepatitis B Virus Infection"

_viruses, 2021, doi:10.3390/v13112216_

Round 1
Reviewer 1 Report
The aim of this manuscript is to analyze large-scale hospital data, focusing on HBV infection and severe liver function, in order to establish if chronic HBV infection, or resolved HBV infection, is a risk factor for a higher ALT elevation (more than 1,000 IU/L).
Even if the manuscript provides an organic overview, with a densely organized structure and based on well-synthetized data, there are aspects to be mentioned, to make the article fully readable. For these reasons, the manuscript requires minor changes.
Please find below an enumerated list of comments on my review of the manuscript:
LINE 41: Hepatitis B Virus infection is a global health problem and a major cause of acute and chronic liver disease. This issue is analyzed by recent studies (see, for reference: Long-term immune protection against HBV: associated factors and determinants – 2021), which also discuss the importance of a long-term immune protection and the immonogenicity of HBV vaccine. This retrospective analysis may benefit from introducing also this issue, in its introductive section .
LINE 238: The decreased risk of HBV-related HCC, due to aspirin daily somministration, is reported also by recent and different studies (see, for reference: Association of Aspirin with Hepatocellular Carcinoma and Liver-Related Mortality – 2021).
MATERIAL AND METHODS:
As regards this section, the methodology design was rigorous and appropriately implemented within the study.
RESULTS and DISCUSSION:
Also these sections are well organized and densely presented, based on well-synthetized data.
In conclusion, this manuscript is densely presented and well organized, based on well-synthetized data. The authors were lucid in their style of writing, making it easy to read and understand the message, portrayed in the manuscript. The article in general is systematic and covers essential literature on this topic, even if requires few changes (as mentioned). I would accept the manuscript, if the comments are addressed properly.
Author Response
Response to reviewer 1
Thank you very much for your invaluable comments.
Response to your comments: “Even if the manuscript provides an organic overview, with a densely organized structure and based on well-synthetized data, there are aspects to be mentioned, to make the article fully readable. For these reasons, the manuscript requires minor changes.”
Thank you very much for your invaluable comments.
I agree with you. Accordingly, we extensively revised our manuscript.
Response to your comments: “LINE 41: Hepatitis B Virus infection is a global health problem and a major cause of acute and chronic liver disease. This issue is analyzed by recent studies (see, for reference: Long-term immune protection against HBV: associated factors and determinants – 2021), which also discuss the importance of a long-term immune protection and the immonogenicity of HBV vaccine. This retrospective analysis may benefit from introducing also this issue, in its introductive section.”
Thank you very much for your invaluable comments.
I agree with you. Accordingly, we added new references [11] and [12], and revised our manuscript as follows.
In lines 57-63 of the revised manuscript,
After 20 years of HBV vaccination, 85–90% of those vaccinated have protective anti-HBs levels [11]. In Japan, although anti-HBV vaccination has been recommended for health care workers, family who lived with HBV-infected individual and newborns from HBV-infected mother, universal vaccination program for children against HBV infection just started at October 2016 [12]. These facts mean that, among almost all Japanese patients, anti-HBs positivity indicates previous HBV infection.
Response to your comments: “LINE 238: The decreased risk of HBV-related HCC, due to aspirin daily somministration, is reported also by recent and different studies (see, for reference: Association of Aspirin with Hepatocellular Carcinoma and Liver-Related Mortality – 2021).”
Thank you very much for your invaluable comments.
I agree with you. Accordingly, we added new reference [23], and revised our manuscript as follows.
In lines 271-275 of the revised manuscript,
Several studies demonstrated that daily aspirin treatment may be associated with a reduced risk of HBV-related HCC occurrence and recurrence, and liver-related mortality [22, 23]. Thus, it is useful to use big data in the analysis HBV infection to establish new findings regarding the association between HBV infection and other diseases [24, 25].

Reviewer 2 Report
Authors presented the retrospective analysis of frequency of patients with HBV-causing ALT elevation using the big data in a single center. There are several concerns as indicated below.
Major point:
1) What is the clinical findings of HBV-causing ALT elevation rates among HBsAg positive patients? The risk factor of HBV-causing ALT elevations among HBsAg positive patients will be valuable. And annual incident rates of HBV-causing ALT elevations among HBsAg positive patients should be presented.
2) Whether or not to take nucleotide or nucleoside analogues among HBsAg positive patients should be shown in Table 1 and Table 2.
3) Backgrounds of highly suspected HBV patients with anti-HBs-positivity and an ALT elevation higher than 1,000 IU/L, who might have HBV reactivation, should be presented. Whether these patients received immunosuppressive treatment or not is important.
4) Results 3.3 showed the frequency of HBV-causing ALT elevations among HBsAg-negative, but positive for anti-HBs/anti-HBc patients, however, the frequency of HBV reactivation in immunosuppressive treatment is important as written in above. How many did patients with positive for anti-HBs/anti-HBc and how long did they receive immunosuppressive treatment? How many patients received nucleotide or nucleoside analogues during immunosuppressive treatment due to the elevation with HBV DNA and could avoided HBV reactivation with ALT elevations?
5) Was sensitivity of HBV markers measurement consistently the same during a-10-year period? Measuring methods of HBV markers should be presented. If the sensitivity of HBsAg changed during a-10-year period, we cannot assess the positivity or negativity of HBsAg in patients with anti-HBc-positivity.
6) How could authors say “These results strongly suggest that attention should be paid to patients with resolved HBV infection in daily clinical practice” in the Discussion section? I think that attention should be paid to patients with resolved HBV infection in immunosuppressive treatment, which is not shown in this paper.
Minor point:
1) I cannot understand the second sentence in first paragraph of Result 3.1. Please explain it easy to understand.
Author Response
Response to reviewer 2
Thank you very much for your invaluable comments.
Response to your major comment 1: “What is the clinical findings of HBV-causing ALT elevation rates among HBsAg positive patients? The risk factor of HBV-causing ALT elevations among HBsAg positive patients will be valuable. And annual incident rates of HBV-causing ALT elevations among HBsAg positive patients should be presented.”
Thank you very much for your invaluable comments.
I agree with you. Accordingly, we revised tables 2-5 and our manuscript as follows.
In lines 237-240 of the revised manuscript,
These results strongly suggest that attention should be paid to patients with resolved HBV infection as well as current HBV infection in daily clinical practice. But even if HBsAg was positive, ~70% of those would not have higher ALT elevation.
In lines 262-264 of the revised manuscript,
…(0.076%) subjects at baseline [6]. In the present study, none had taken nucleos(t)ide analogues at higher ALT elevations than 1,000 IU/L among HBsAg-positive group (Table 2) or HBsAg-negative and anti-HBs and/or anti-HBc-positive group (Table 4).
In lines 281-295 of the revised manuscript,
HBV reactivation associated with the use of immunosuppressive treatment remains one of the major causes of liver related deaths in Asia-Pacific region, where HBV infection is endemic [26]. In the present study, we observed HBV reactivation associated with the use of immunosuppressive treatment in two patients (Tables 2 and 4). Attention should be paid to patients with resolved HBV infection in immunosuppressive treatment (Table 4). It seems important to screen all patients for HBV infection before the beginning of immunosuppressive treatment, and administer pre-emptive nucleos(t)ide analogues to some patients with a risk of HBV reactivation [26-28].
There are several limitations of this study. In the present study, we observed ALT elevations higher than 1,000 IU/L among HBsAg-positive patients, compared with other groups. As it has been difficult to perform the continuous observation for each patient for long time in only one hospital, we could not calculate annual incident rates of HBV-causing ALT elevations among each group. The sensitivity of the measurement of HBV markers, including HBsAg and HBV DNA, has been changed [29]. But our study demonstrated the real-world data in our hospital [30, 31]. Further study may be needed.
Response to your major comment 2: “Whether or not to take nucleotide or nucleoside analogues among HBsAg positive patients should be shown in Table 1 and Table 2.”
Thank you very much for your invaluable comments.
I agree with you, to some extent. Accordingly, we revised table 2, however, as our study is a big-data analysis, we could not revise table 1, and will need to perform additional analysis in future study.
Response to your major comment 3: “Backgrounds of highly suspected HBV patients with anti-HBs-positivity and an ALT elevation higher than 1,000 IU/L, who might have HBV reactivation, should be presented. Whether these patients received immunosuppressive treatment or not is important.”
Thank you very much for your invaluable comments.
I agree with you. Only one patient (Case 28 in table 4) had received rituximab- cyclophospha-mide/hydroxydaunorubicin/oncovin/prednisolone (R-CHOP) for 0.5 years. Accordingly, we revised new table 4 and our manuscript as follows.
In lines 283-286 of the revised manuscript,
…is endemic [26]. In the present study, we observed HBV reactivation associated with the use of immunosuppressive treatment in two patients (Tables 2 and 4). Attention should be paid to patients with resolved HBV infection in immunosuppressive treatment (Table 4). It seems important to screen all patients for HBV infection before the…
Response to your major comment 4: “Results 3.3 showed the frequency of HBV-causing ALT elevations among HBsAg-negative, but positive for anti-HBs/anti-HBc patients, however, the frequency of HBV reactivation in immunosuppressive treatment is important as written in above. How many did patients with positive for anti-HBs/anti-HBc and how long did they receive immunosuppressive treatment? How many patients received nucleotide or nucleoside analogues during immunosuppressive treatment due to the elevation with HBV DNA and could avoided HBV reactivation with ALT elevations?”
Thank you very much for your invaluable comments. I agree with you, to some extent. Accordingly, we revised table 4, however, as our study is a big-data analysis, we could not unfortunately show how many patients with positive for anti-HBs/anti-HBc did and how long they received immunosuppressive treatment and we need to perform another analysis in future study. But only one of 7 patients (in table 4) had received rituximab- cyclophospha-mide/hydroxydaunorubicin/oncovin/prednisolone (R-CHOP) for 0.5 years. Accordingly, we revised new table 4 and our manuscript as follows.
In lines 283-286 of the revised manuscript,
…is endemic [26]. In the present study, we observed HBV reactivation associated with the use of immunosuppressive treatment in two patients (Tables 2 and 4). Attention should be paid to patients with resolved HBV infection in immunosuppressive treatment (Table 4). It seems important to screen all patients for HBV infection before the…
Response to your major comment 5: “Was sensitivity of HBV markers measurement consistently the same during a-10-year period? Measuring methods of HBV markers should be presented. If the sensitivity of HBsAg changed during a-10-year period, we cannot assess the positivity or negativity of HBsAg in patients with anti-HBc-positivity.”
Thank you very much for your invaluable comments.
I agree with you. Accordingly, we add the limitation of our study and revised our manuscript as follows.
In lines 289-295 of the revised manuscript,
There are several limitations of this study. In the present study, we observed ALT elevations higher than 1,000 IU/L among HBsAg-positive patients, compared with other groups. As it has been difficult to perform the continuous observation for each patient for long time in only one hospital, we could not calculate annual incident rates of HBV-causing ALT elevations among each group. The sensitivity of the measurement of HBV markers, including HBsAg and HBV DNA, has been changed [29]. But our study demonstrated the real-world data in our hospital [30, 31]. Further study may be needed.
Response to your major comment 6: “How could authors say “These results strongly suggest that attention should be paid to patients with resolved HBV infection in daily clinical practice” in the Discussion section? I think that attention should be paid to patients with resolved HBV infection in immunosuppressive treatment, which is not shown in this paper.”
Thank you very much for your invaluable comments.
I agree with you. Accordingly, we revised our manuscript as follows.
In lines 237-240 of the revised manuscript,
…These results strongly suggest that attention should be paid to patients with resolved HBV infection as well as current HBV infection in daily clinical practice. But even if HBsAg was positive, ~70% of those would not have higher ALT elevation.
In lines 281-288 of the revised manuscript,
HBV reactivation associated with the use of immunosuppressive treatment remains one of the major causes of liver related deaths in Asia-Pacific region, where HBV infection is endemic [26]. In the present study, we observed HBV reactivation associated with the use of immunosuppressive treatment in two patients (Tables 2 and 4). Attention should be paid to patients with resolved HBV infection in immunosuppressive treatment (Table 4). It seems important to screen all patients for HBV infection before the beginning of immunosuppressive treatment, and administer pre-emptive nucleos(t)ide analogues to some patients with a risk of HBV reactivation [26-28].
Response to your minor comment: “I cannot understand the second sentence in first paragraph of Result 3.1. Please explain it easy to understand.”
Thank you very much for your invaluable comments.
I agree with you. Accordingly, we revised our manuscript as follows.
In lines 131-135 of the revised manuscript,
First, we examined how often and how frequent the HBV and HCV viral markers were measured in 73,634 and 72,700 individuals, respectively. In total, 811 (1.1%) of 73,379, 2,369 (25.7%) of 9,214, 2,128 (28.6%) of 7,428, 78 (10.0%) of 781, 463 (60.0%) of 772 and 2,959 (4.1%) of 72,700 patients were positive for HBsAg, anti-HBs, anti-HBc and anti-HCV, respectively.

Reviewer 3 Report
In the study entitled "The use of electronic medical records-based big-data informatics to describe ALT elevations higher than 1,000 IU/L in patients with or without hepatitis B virus infection" by Amano et al., the authors used big data from their hospital to predict the outcome of patients with ALT elevation. They propose to carefully check HBsAg negative patients with ALT higher than 1000 IU/L for a risk of rebound of HBV DNA. Although the method is good with the usage of a significant number of patient data, the study is poorly written, which substantially decreases the soundness of the study. The authors need to make an effort in properly describing and interpreting the data to clarify the point they want to make.
Other comments:
- The authors should provide with the reference of the kit they used to detect HBV DNA and a reference justifying to state that patients with HBV DNA content lower than 2.1 logIU/mL are considered HBV DNA negative
- In Table 2 and 3, all the patients should be shown even the ones they consider non-HBV.
Author Response
Response to reviewer 3
Thank you very much for your invaluable comments.
Response to your comments: “Although the method is good with the usage of a significant number of patient data, the study is poorly written, which substantially decreases the soundness of the study. The authors need to make an effort in properly describing and interpreting the data to clarify the point they want to make.”
Thank you very much for your invaluable comments.
I agree with you. Accordingly, we extensively revised our manuscript.
Response to your other comments: “The authors should provide with the reference of the kit they used to detect HBV DNA and a reference justifying to state that patients with HBV DNA content lower than 2.1 logIU/mL are considered HBV DNA negative”
Thank you very much for your invaluable comments.
I agree with you. Accordingly, we revised our manuscript as follows.
In lines 106-113 of the revised manuscript,
2.3. Measurement of HBV DNA levels
The HBV DNA level was measured by a commercially available real-time PCR-based assay: transcription-mediated amplification (TMA) assay, COBAS Amplicor HBV Monitor assay, or COBAS TaqMan (Roche Diagnostics, Branchburg, NJ, USA) ) [16]. We converted the HBV DNA levels as following formula: (log IU/mL) = (log copies/mL)-0.76 (http://www.jsh.or.jp/member/archives/21, accessed on 11/1/2016). HBV DNA equal to or more than 2.1 log IU/mL was defined as HBV DNA detectable, and HBV DNA less than 2.1 log IU/mL was defined as HBV DNA undetectable in the present study.
Response to your other comments: “In Table 2 and 3, all the patients should be shown even the ones they consider non-HBV.”
Thank you very much for your invaluable comments.
I agree with you, to some extent. Accordingly, we made a new table 3, showing “non-hepatitis B virus (HBV) suspected patients who had higher alanine transaminase (ALT) el-evations than 1,000 IU/L among the HBV surface antigen (HBsAg)-positive group,” and a new table 5, showing “various type of liver diseases causing non-hepatitis B virus (HBV) related higher alanine trans-aminase (ALT) elevations than 1,000 IU/L among the HBV surface antigen (HBsAg)-negative but anti-HBV surface antibody (anti-HBs)/anti-HBV core antibody (anti-HBc)-positive group and tri-ple negative group” because these groups included many patients.

Round 2
Reviewer 2 Report
Thank you for addressing my concern.
Reviewer 3 Report
The authors properly addressed all my comments.